# Cerebellum’s Contribution to Attention, Executive Functions and Timing: Psychophysiological Evidence from Event-Related Potentials

**DOI:** 10.3390/brainsci13121683

**Published:** 2023-12-07

**Authors:** Daniela Mannarelli, Caterina Pauletti, Paolo Missori, Carlo Trompetto, Filippo Cotellessa, Francesco Fattapposta, Antonio Currà

**Affiliations:** 1Department of Human Neurosciences, Sapienza University of Rome, Viale dell’Università 30, 00185 Rome, Italy; daniela.mannarelli@uniroma1.it (D.M.); caterina.pauletti@uniroma1.it (C.P.); missorp@yahoo.com (P.M.); francesco.fattapposta@uniroma1.it (F.F.); 2Department of Neuroscience, Rehabilitation, Ophthalmology, Genetics, Maternal and Child Health, University of Genoa, 16132 Genoa, Italy; ctrompetto@neurologia.unige.it (C.T.); filippo_cotellessa@hotmail.it (F.C.); 3IRCCS Ospedale Policlinico San Martino, Division of Neurorehabilitation, Department of Neuroscience, 16132 Genoa, Italy; 4Academic Neurology Unit, Department of Medico-Surgical Sciences and Biotechnologies, Sapienza University of Rome, 04019 Terracina, Italy

**Keywords:** cerebellum, cognition, ERPs, attention, timing, neuromodulation, cerebello-cerebral networks, executive functions

## Abstract

Since 1998, when Schmahmann first proposed the concept of the “cognitive affective syndrome” that linked cerebellar damage to cognitive and emotional impairments, a substantial body of literature has emerged. Anatomical, neurophysiological, and functional neuroimaging data suggest that the cerebellum contributes to cognitive functions through specific cerebral–cerebellar connections organized in a series of parallel loops. The aim of this paper is to review the current findings on the involvement of the cerebellum in selective cognitive functions, using a psychophysiological perspective with event-related potentials (ERPs), alone or in combination with non-invasive brain stimulation techniques. ERPs represent a very informative method of monitoring cognitive functioning online and have the potential to serve as valuable biomarkers of brain dysfunction that is undetected by other traditional clinical tools. This review will focus on the data on attention, executive functions, and time processing obtained in healthy subjects and patients with varying clinical conditions, thus confirming the role of ERPs in understanding the role of the cerebellum in cognition and exploring the potential diagnostic and therapeutic implications of ERP-based assessments in patients.

## 1. Introduction

The cerebellum is widely known to be associated with motor control [1], but several studies of neuroanatomy, neuroimaging, neuropsychology, and non-invasive brain stimulation (NIBS), have implicated the cerebellum in specific cognitive domains [2,3,4,5] such as attention [6,7,8,9,10,11], executive functioning [12], temporal representation [13], language processing [14,15], visuospatial cognition and personality changes that may result in a general decline in intellectual functioning (so-called “cerebellar cognitive affective syndrome”, CCAS, [16]).

The cerebellar contribution to cognition is mediated by specific cerebral–cerebellar crossed connections [17], organized in parallel loops, with a prominent role played by the cortico-ponto-cerebello-dentato-thalamo-cortical pathway [2,18,19]. Output from the cerebellar cortex through the dentate nucleus reaches the cerebral cortex diffusely [20]. The main part of this output is for higher-level prefrontal and parietal cortices, but temporal areas also have reciprocal connections with the cerebellum mediated via the pons [21,22,23]. Cerebellar–cortical connections also depart from the interposed nucleus and the fastigial nucleus, and (through thalamic interconnections) reach brain areas involved in emotional control such as the amygdala, hippocampus and middle temporal gyrus [24,25].

Neuroimaging studies, including those adopting resting-state functional connectivity, have confirmed dense connections between the movement-related cerebellar areas (lobule I, IV, V and anterior area of lobule VI) and the contralateral primary motor and primary somatosensory cortex [26,27,28,29,30], but also strong connectivity between cerebellar regions, brain associative cortices and systems of interconnected neurons such as the salience network and the default-mode network [29,30,31,32]. Finally, right lobules VI and Crus I are involved in language processing, and left lobule VI is involved in visuospatial processing [3].

A considerable effort has been made to understand the cerebellar contribution to cognition, yielding interesting results from many neuroimaging studies as well as neuropsychological studies, including those on psychiatric and developmental disorders [14,15,33,34,35,36].

Physiologically, the main cerebellar function is the inhibitory firing toward the brain areas. This is called cerebellar brain inhibition tone [37,38], and it influences both motor and cognitive circuits via a synaptic relay in the ventral-lateral thalamus [39,40]. The dentate-thalamus pathway influences the functioning of parietal and the frontal cortices [39] involved in attention and in executive control. The cerebellum modulates the levels of activation or inhibition of cerebral cortical areas, thus acting as a “coordinator” that indirectly influences the cognitive processes. A specific cerebellar function is predictive coding, that is, the ability to make specific sensory predictions according to the anticipatory or feed-forward model [4,12]. Predictive coding works by evaluating temporal patterns of stimuli, errors or conflicting signals with the aim of improving the reliability of future predictions and producing online changes in behavior, thereby influencing inhibitory control.

The purpose of this review is to examine the use of event-related potential (ERP) techniques in the investigation of cerebellar contributions to cognitive functioning not related to motor preparation, control or execution.

When a stimulus, or a combination of stimuli, of any sensory modality (auditory, visual, somatosensory) activates a higher-level neural process, thus becoming a so-called “event,” the resulting flow of brain information can be traced by the ongoing electroencephalogram (EEG) recorded from the scalp [41]. A waveform representative of the changes in brain voltage related to the underlying neural process can be obtained by averaging the EEG activity over multiple trials. This waveform is called event-related potential (see Figure 1, Adapted from Woodman, 2010 [42]) and it consists of various components. Depending on the experimental conditions, ERP components can be related to events such as predicting errors (mismatch negativity (MMN), error-related negativity (ERN) and error-related positivity (Pe)), switching attention to task-irrelevant novel stimuli (P3a), categorizing target stimuli (P3b) [43], establishing the temporal contingency between stimuli (contingent negative variation (CNV)) [44] and so on, up to imagery processes [45]. ERP components are described by their polarity (positive or negative), their amplitude (reflecting how much neural activity is allocated to the current cognitive function), their latency (signaling the time at which the neural activity occurs) and their scalp distribution (the superficial projection of potentials generated by cortical structures) [46].

ERPs represent a reliable and non-invasive method to acquire real-time knowledge about how the human brain processes information with an exceptional temporal resolution, in healthy subjects and patients with neurological conditions.

The ability to track selective cognitive functions and monitor them through quantitative parameters such as amplitude, latency and distribution provides ERPs with a clinical value similar to that of short-latency evoked potentials, although the neural generators of the late components that form ERPs are much more complex of those that give rise to the short-latency evoked components [46].

## 2. Cerebellum and Attention

Psychophysiological studies of adult patients with acquired cerebellar lesions provided evidence for the cerebellum’s role in various aspects of attention [47,48,49,50]. We reported a reduction in P3 amplitude evoked by a classic auditory oddball paradigm in a man with a left posterior cerebellar ischemic stroke [49]. The P3 improved after 4 weeks. These observations align with a previous case report that described P3 alterations (also evoked by auditory stimuli) in a patient with a large cerebellar lesion [47]. The P3 changes observed in these patients likely reflected attentional dysfunction, thus supporting the contribution of the cerebellum to the attentional processing of the stimulus. Our observation that P3b amplitude restores after 4 weeks from stroke, which paralleled the recovery of the attentional cognitive functioning, suggests a possible role of the P3 component as a neurophysiological marker of functional cerebellar recovery. To confirm that the cerebellum contributes to attention, early and late ERP components (early posterior negativity (EPN) and late positive potential (LPP)) to highly and poorly emotionally arousing pictures (pleasant, unpleasant and neutral pictures of the International Affective Picture System), with and without competing attentional tasks, were recorded in a patient with an ischemic cerebellar infarct [50]. The EPN response to highly arousing emotional cues in the competing visual attention condition was absent, whereas the LPP response to highly arousing emotional cues showed augmentation over frontal areas. This pattern of ERP findings suggested specific neural dysfunction associated with emotional–behavioral disturbances following cerebellar lesions, pointing toward a role of the cerebellum in supporting emotional attention. Another study in patients with cerebellar lesions used a N100/N1 suppression paradigm combining self-initiated with externally triggered auditory stimuli, to explore the ability to allocate attention to stimuli related to the prediction of an expected sensory information [48]. Generally, the N1 is suppressed when sensory information matches the prediction of an expected stimulus so that the brain activity directed to the actual input is reduced. Self-initiated sensory stimuli are highly predictable, while externally triggered stimuli lead to an increased processing activity as external sensations may provide new and important information. Patients lacked N1 suppression in response to self-initiated sounds, thus suggesting that the cerebellum is essential for generating internal forward predictions [48].

Other interesting insights arise from studies on genetically determined cerebellar ataxias (CA) in which cognitive decline is a common non-motor feature. Using behavioral tests with concomitant EEG recordings, sensory predictive coding processes and response adaptations were examined [51]. Sensory prediction coding was tested with an auditory distraction paradigm and error-related behavioral adaptations were tested with a visual flanker task. Many ERPs were measured, including the P3a for the orientation of attention, the N2 and the ERN for the cognitive adaptation processes/consequences of response errors, the Pe for error awareness, the MMN for sensory predictive coding and automatic involuntary attention and the reorientation negativity (RON) for reorientation after unexpected events. ERPs related to voluntary cognitive processing, including attentional switching (P3a, RON) and error awareness (Pe), were abnormal in CA patients, whereas attentional functioning that generates internal automatic forward processes resulted largely intact (MMN, ERN/N2).

Attentional abnormalities were also observed in patients with spinocerebellar ataxia 2 (SCA2), who showed abnormal visual/cognitive processing measured by visually evoked potentials (VEPs) and P3 components elicited by a visual oddball task [52]. Similarly to what was observed with auditory stimuli, recordings of visual paradigms showed a lower P3 amplitude and prolonged P3 latency [53], thus suggesting attentional, discriminative and working memory abnormalities in these patients. Preclinical SCA2 carriers exhibited less severe but significant prolongation of P3 latencies. Overall, these findings provide evidence supporting the cerebellar involvement in attention and memory, and they show that psychophysiological measures may act as the biomarkers of the cognitive decline in SCA [53].

A recent P3 study evoked by a dual-task emotion perception task that adopted angry, happy and neutral facial expressions explored the presence of cognitive deficits in Type I Chiari malformation. This impairment is thought to be related directly to compression dynamics at the cervico-medullary junction and indirectly to long-term chronic pain experienced by patients [54]. Although patients had slower response times than normal, they did not differ from controls in the ability to allocate attentional resources. However, patients had an increased frontal representation of the P3 amplitude, reflecting compensatory neural recruitment.

Interestingly, the cerebellum emerged as one of the key dysfunctional brain cognitive nodes in specific pathological populations with neuropsychiatric diseases [55,56,57,58]. While investigating the neural basis of the abnormalities in eye gaze processing with source imaging analysis in patients with attention deficit hyperactivity disorder (ADHD), the P200 wave at the left/midline cerebellum was found to be reduced [59]. Based on the growing evidence of the cerebellar involvement in emotion recognition, theory of mind and empathy [60,61], the reduced cerebellar activity was thought to lead patients to dysfunctionally integrate social inputs to attentional executive tasks. In a double-blind placebo-controlled study, the efficacy of 3-week prefrontal-excitatory and cerebellar-inhibitory transcranial direct current stimulation (tDCS) on neurocognitive functioning was explored in patients with the euthymic bipolar disorder [62]. They showed improved executive functioning (trail making test—B) and visuospatial memory (Rey complex figure test delay recall) after an active tDCS session, accompanied by a decrease in the P3b latency. These findings suggest that cerebellar tDCS improves the attentional brain processing stream, and indicates that the prefrontal cortex, cerebellum and prefrontal–thalamic–cerebellar circuitry are implicated in cognitive processes, including those reported as dysfunctional in these patients with the euthymic bipolar disorder. Concomitant prefrontal-excitatory and cerebellar-inhibitory tDCS may represent a useful rehabilitative tool for better neurocognitive performance.

Cerebellar involvement in attention functioning was also explored in healthy subjects. The role in pre-attentive automatic change detection processes was suggested by ERP studies showing changes in N1 and MMN components after cerebellar modulation with non-invasive brain stimulation (NIBS) techniques [63,64]. Investigating the effects of tDCS delivered over the left cerebellar hemisphere in cathodal (inhibitory), anodal (excitatory) and sham sessions during a P3 Novelty auditory task, cathodal cerebellar tDCS alone reduced the amplitude of the N1, N2 and P3 components for both the target and novel stimuli, and shortened the N1 latency evoked by each stimulus (target, novel and standard) [65]. These ERP changes show that the cerebellum operates in different stages of the attentional processing of the stimulus, from the automatic involuntary detection (N1) and the early phase of the attention switching (N2) to the attentive discrimination of the stimuli (P300), by regulating the activation and inhibition levels of the brain cortical areas involved in attentional networks. The role of the cerebellum in the functioning of the attention networks was also demonstrated by using the attention network task combined with tDCS [66]. In this study, we reported a selective reduction in the efficiency of the executive network after cerebellar inhibition, specifically related to the ability to process complex stimuli in which conflict signals or errors are present.

Combining functional neuroimaging (fMRI) and ERP data during a modified auditory oddball paradigm for the elicitation of P3 components related to attention and working memory cognitive processes as well as the activation of fronto-parietal areas and the cerebellum was found in healthy children between the ages of 11 and 16 (Figure 2) [67].

Lastly, when examining the neural mechanisms of emotional word processing in bilinguals with an ERP–fMRI registration, emotional processing activated a rapid and automatic attentional orienting response during left cerebellum activation [68].

## 3. Cerebellum and Executive Functioning

The term “executive functions” refers to the ability to coordinate different cognitive tasks to obtain specific goals [69]. It consists of various cognitive abilities, such as working memory, problem solving, cognitive flexibility, preparation and inhibitory control, necessary to plan and direct goal-oriented behavior. The prefrontal cortex is crucial in maintaining executive control, also thanks to a strong fronto-cerebellar connectivity, consisting of closed cortico-cerebellar loops in which the dorsolateral prefrontal cortex connects to the cerebellum via pontine nuclei, and the cerebellum sends projections back to the prefrontal cortex via the dentate nucleus and thalamus [70,71].

Cerebellar involvement in executive functioning has been documented by experimental ERP studies, primarily in patients with movement disorders, such as Parkinson’s disease.

To investigate the preparation of self-paced and externally cued movements in patients with Parkinson’s disease and essential tremors, motor-related potentials (MRP), such as the Bereitschaftpotential (BP) and the contingent negative variation (CNV), were recorded [72]. BP is recorded over the motor cortices and is best represented contralateral to the finger performing self-paced movements; CNV is recorded over fronto-parietal cortices bilaterally and reflects the attentional anticipation and motor preparation to externally cued movements [73]. Motor preparation goes under the control of distinct cortico-thalamo-cortical circuits according to various motor parameters including the type of cueing. The pattern of findings in BP and CNV in the two group of patients (having predominant basal ganglia vs. predominant cerebellar dysfunction) showed that the cortico-basal ganglia–thalamocortical circuit prepared both self-paced and externally cued movements, whereas the cerebello-dentato-thalamocortical pathway prepared only self-paced movements. This observation is especially interesting because many neurophysiological studies have shown that this pathway is involved in executing externally cued movements that, indeed, are relatively spared in patients with Parkinson’s disease [74]. A reasonable explanation for cerebellum involvement in preparing self-paced movements is that, during motor preparation, subjects were asked to determine the timing of the self-paced task, and cerebellum is known to manage the timing of movement preparation.

Some insights related to cerebellar involvement in cognitive flexibility were obtained from psychophysiological data in drug abusers [75]. By studying a conflict task designed to elicit a slow EEG potential (SP, a P3-like potential that emerges approximately 500 ms after stimulus onset) and using these data to localize the neural substrates of response dysregulation, the SP amplitude showed a normal spatial conflict effect for opioid-dependent and non-opioid-dependent subjects, but not for cocaine-dependent patients. Correlational analysis showed that abnormal SP was not related to quantity, frequency or recent cocaine use, but depended on comorbid alcohol use. A neuroanatomical localization algorithm applied to SP data showed that comorbid alcohol use disrupted normal task-related activation of the anterior cingulate, prefrontal cortex and cerebellum.

In one patient with a large cerebellar lesion, the P300 component was prolonged, reduced and changed in morphology, with two distinct peaks not normally elicited by the classical auditory oddball P300 paradigm. That patient performed poorly in neuropsychological tests, with difficulties in planning, abstract reasoning, set-shifting and perseveration [47]. Executive dysfunction was considered the consequence of changes in cerebello-frontal circuitry, since P300 is generated by cortical (especially prefrontal) and subcortical areas [76,77].

In a recent study aimed to evaluate the executive inhibitory control in cognitively intact APOE4 non-carrier elders, the EEG source analysis revealed greater cerebellar activity during the P300 time window in a stop-signal task that used visual stimuli. This activity was considered compensatory and was absent in healthy elderly APOE carriers, indicating that APOE carriers, even when asymptomatic, lack cerebellar compensatory mechanisms [78].

The cerebellar contribution to inhibitory executive control emerges from a study in which the profile of cognitive impairment was assessed in patients with cerebellar cortical atrophy [79]. To assess attentional performance and the ability to control a motor response, subjects were subjected to a comprehensive battery of neuropsychological tests along with a conventional auditory oddball task and a continuous performance task, i.e., execute (“Go”) or inhibit a motor reaction (“No Go”). Baseline-independent measures (global field power (GFP)) were determined, and low-resolution brain electromagnetic tomography (LORETA) was used to calculate the three-dimensional intracerebral distribution of electrical activity of the P3 component of Go and NoGo responses. Patients had prolonged GFP peak latency and attenuated GFP peak specifically in the NoGo condition, which is associated with LORETA evidence of low frontal NoGo P3 source activation. This pattern of findings suggests that cerebellar degeneration contributes to frontal executive dysfunction by impairing the inhibitory executive system.

A visual Go/NoGo task with concomitant LORETA was used to investigate P2/P3 potentials and brain generators in children with ADHD [80]. They had reduced Go and NoGo-P3 components due to a decreased contribution of frontal areas and dorsal ACC, respectively. Concomitantly, the increased NoGo-P2 amplitude was the result of the decreased contribution of the dorsolateral prefrontal cortex, the insula and the cerebellum. These findings suggest the fronto-cerebellar involvement in the automatic feature of inhibition processes.

Inhibitory control was analyzed in healthy subjects who underwent an auditory Go/NoGo task before and after cathodal and sham cerebellar tDCS in separate sessions [81]. Cathodal but not sham tDCS prolonged and reduced the N2-NoGo potential, indicating that tDCS-induced cerebellar inhibition worsened the ability to allocate attentional resources to stimuli containing hostile information and consequently impaired inhibitory control. Changes in the N2-NoGo potential suggest that the cerebellum contributes to regulating attentional mechanisms of stimulus orienting and inhibitory control both directly by predicting errors or error-related behavior and indirectly by controlling the functioning of the cortical areas involved in signal perception of conflict and the basal ganglia involved in the inhibitory control of movement.

The cerebellar role in executive functioning was also investigated in healthy subjects using repetitive transcranial magnetic stimulation technique during a visual 2-back task commonly used to explore working memory processes [82]. Further, 5 Hz and 20 Hz stimulation on the Crus II region of the cerebellum, respectively, increased N170 amplitude in the prefrontal areas and the P300 amplitude in the prefrontal and parietal sites. Cerebellar excitatory rTMS proved effective in modifying cognitive markers of working memory, bringing further evidence of the cerebellum’s contribution to cognition.

Various tasks using visual stimuli were delivered to musically trained and untrained normal young individuals to investigate the cerebellar involvement in executive functioning [83]. Neuropsychological, psychophysiological and fMRI evaluations proved “not normal” in musically trained young individuals. Specifically, the P3b component to incongruent target stimuli was more posteriorly distributed on the scalp, similarly to the adult P3b response, a finding thought to reflect an early maturity of updating and working memory functions related to target processing. During set-shifting tasks, the fMRI data showed less activity in frontal, parietal and occipital areas of the dorsal attention network and in the cerebellum, indicating that musically trained young individuals have a more efficient recruitment of neural resources after childhood.

## 4. Cerebellum and Timing Processing

As previously suggested by Purzner et al. [72], the cerebellum is also engaged in timing control of movement preparation. Cerebellar involvement in time perception has its roots in long-established clinical observations that motor coordination can be severely disrupted by cerebellar injury [84]. With the loss of precise timing information, motor acts and internal cognitive processes may no longer be appropriately selected and sequenced at a fine level. Thus, motorically, individuals may become less coordinated, and, cognitively, they may exhibit the so-called “dysmetria of thought” with problems in task shifting and other forms of executive cognitive control. The cerebellum is considered a critical substrate for the perceptual timing of single intervals [13] with a specific role in the discrimination of sub-second time range that is a more automatic system closely linked with motor circuits [85,86].

A hierarchical set of timing tasks has been created to examine the cerebellar role in perceptual timing. Specifically, two different levels of time measures were individuated: a more basic one related to the duration discrimination of single stimuli or intervals and a more complex one related to the discrimination of the rhythmic patterns of a temporal representation.

Few ERP studies investigated the cerebellar role in time control. An auditory mismatch paradigm was analyzed in patients with bilateral cerebellar degeneration to evaluate sensory prediction of temporal regularity [87]. Patients had a MMN of normal amplitude but prolonged latency for stimulus duration deviants alone, but not for pitch and location deviants. This finding reflects an impairment at the early stage of auditory processing (100–200 ms), which is the automatic phase of cognitive processing for temporal estimation, and provides support for a cerebellar contribution to the automatic, pre-attentive duration estimation of the stimuli.

A later study provided evidence that the voluntary processing of the temporal structure of events can also be influenced by the cerebellum [88]. During a P300 auditory oddball paradigm, cerebellar patients and healthy controls displayed a normally increased N2b response to deviant tones regardless of the temporal context. However, whereas healthy controls expectedly enhanced the P3b response to deviant tones in temporally regular sequences, patients unexpectedly decreased the response. These results indicate that structural damage to the cerebellum affects the predictive adaptation to the temporal structure of events and the updating of a mental model of the environment under voluntary attention. In support of this view, by combining magnetoencephalographic (MEG) and EEG recordings in normal subjects during the performance of intermittent electrical stimulation of the finger with random stimulus omissions, the violations of temporal expectancies in the somatosensory domain produced a localized physiological signal to the cerebellum [89].

Most recently, to explore how the cerebellum estimates the duration of time intervals, a CNV paradigm elicited by S1-S2 motor tasks was analyzed in healthy subjects before and after cathodal and sham cerebellar tDCS (Figure 3) [90]. The CNV task consisted of a duration discrimination task in which subjects had to determine whether the duration of a probe interval trial was shorter (800 ms), longer (1600 ms) or equal to the target interval of 1200 ms. CNV amplitude decreased only after cathodal tDCS for short and target interval trials, but not for the long interval trials, suggesting that cerebellar inhibition induced by tDCS impaired the perception of short timing intervals. These data point toward a selective involvement of the cerebellum in second and sub-second timing control, as is also shown by ERP findings in rat studies [91,92].

## 5. Limitations and Future Directions

We have presented psychophysiological evidence from ERP studies on the cerebellum’s contribution to attention, executive functions and timing. The ERP technique has some limitations. First, ERP components may vary inter-individually in amplitude, due to differences in cortical folding patterns, or skull thickness [93]. The cognitive processes that ERPs explore are complex in nature, and more than one function at a time could be engaged during many tasks. Therefore, the psychophysiological components may be evoked by an overlap of various functions rather than a selective process [94], and the cerebellum’s contribution to a specific cognitive function may be difficult to unravel. As opposed to the high temporal resolution, the scarce spatial resolution offered by ERPs may be considered detrimental; however, analysis algorithms, such as LORETA, provide reliable information regarding the sources of activity.

Some of the studies reviewed were conducted on small-sized samples (Table 1), thus raising possible issues of statistical power. However, this was unavoidable for case reports and studies in patients with rare neurological conditions. The groups studied, the tasks performed, the sensory modalities of delivered stimuli, the recording techniques and the analysis procedures differed between studies; thus, the conclusions are not easily comparable. On the other hand, across the many small differences found using the research methods, the role of the cerebellum in the investigated cognitive process emerges anyway. Thus, further studies are needed to confirm the reviewed findings that include larger population samples and more standardized protocols.

However, using ERPs as a research tool has numerous advantages. ERPs are non-invasive, making them safe and well tolerated by participants. They are relatively low cost compared to neuroimaging techniques; they have lower technological requirements than other research methods, which makes them accessible in various clinical and investigational contexts. Overall, ERPs provide a valuable and versatile tool for studying cognitive neural processing in the brain, possibly serving as a biomarker of brain dysfunctions not detected by traditional clinical tools [93].

The combination of ERP techniques with NIBS has proven to be extremely useful for testing specific cognitive functions in healthy subjects and for monitoring therapeutic benefit in neurological conditions. By inducing transient changes in local neural networks, NIBS can help distinguish the contribution of brain structures to individual stages of cognitive processes, as reflected by ERP components [95]. Finally, ERPs are particularly well suited to monitor the longer-lasting effect of stimulation and to test whether this effect may be beneficial in neurological conditions, including cerebellar involvement.

## 6. Conclusions

The psychophysiological evidence reviewed here indicates that ERPs detect the contribution of the cerebellum to cognition. The cerebellum coordinates the correct development of cognitive processes by regulating highly organized cerebro-cerebellar neural networks, and cerebellar dysfunction impairs cognitive processes such as attention, executive functions and timing.

Attention-related ERP components, especially P3, are reduced in neurological conditions characterized by cerebellar damage (including stroke, SCA, ADHD), and in subjects with virtual cerebellar lesions induced by NIBS. Executive functions-related ERPs have prolonged latency (for instance, P3, NoGo GFPm, NoGo N2) following cerebellar damage. Timing-related ERPs reveal that both degenerative and virtual cerebellar lesions prolong MMN latency or reduce CNV amplitude.

Despite their agreement in identifying psychophysiological abnormalities, ERP studies exploring the contribution of the cerebellum to cognition are scarce. This highlights the need for further research to confirm the reviewed findings, to gain a deeper understanding of the cerebellar cognitive mechanism and to explore the potential diagnostic and therapeutic implications of ERP-based assessments in patients with cerebellar disorders.

## Figures and Tables

**Figure 1 brainsci-13-01683-f001:**
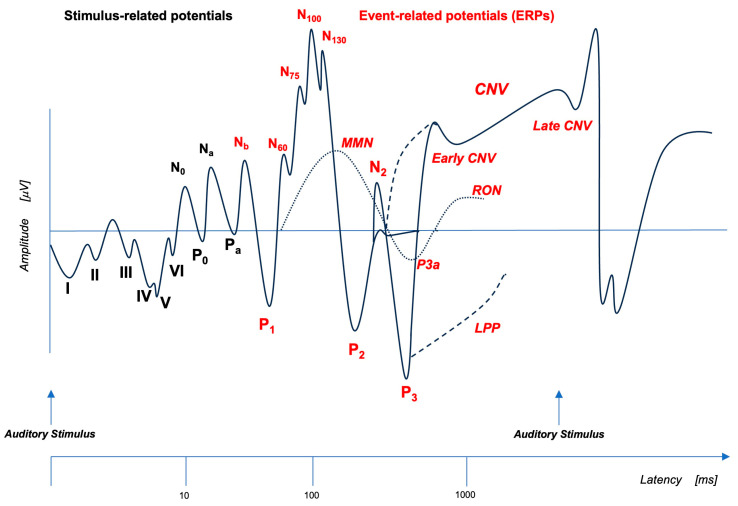
Event-related potentials (ERPs) represent a measure of cortical electric activity evoked during a cognitive task and recorded from scalp with EEG. Here, a schematic representation of auditory ERPs detected after EEG signal averaging analysis is shown in logarithmic temporal sequence. Early responses to auditory stimuli (AEP) are depicted with black letters. Long latency waves, time-locked to the auditory event (ERPs), are depicted with red letters. Each component is described by a letter that indicates its polarity (P: positive and N: negative) and by a number/letter that indicates its position in the sequence or a number that indicates its latency. CNV: contingent negative variation; LPP: late positive potential; MMN: mismatch negativity; RON: reorienting negativity (Adapted from Woodman, 2010 [42]).

**Figure 2 brainsci-13-01683-f002:**
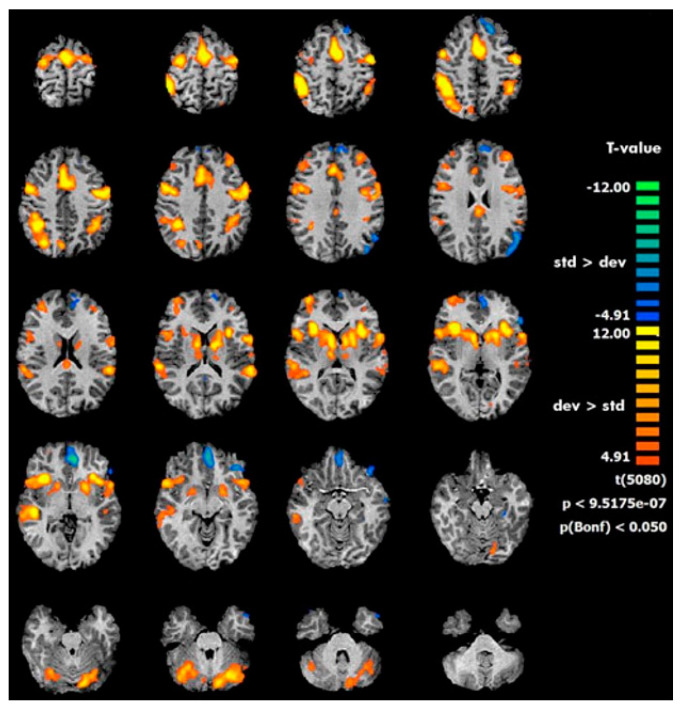
fMRI activations during oddball paradigm in pediatric population (from Rusiniak et al., 2013) [67].

**Figure 3 brainsci-13-01683-f003:**
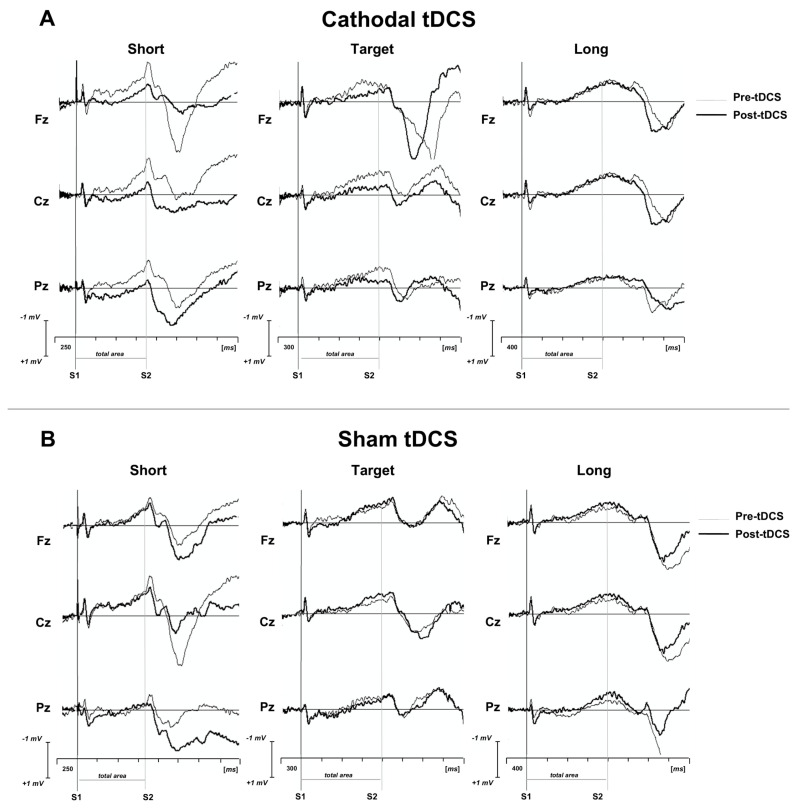
Grand average CNV traces superimposed for pre- and post-tDCS for both (**A**) cathodal and (**B**) sham conditions for short, target and long time intervals. Short trial was 800 ms; target trial was 1200 ms; long trial was 1600 ms. S1: warning auditory stimulus (frequency: 1000 Hz; duration: 50 ms; intensity: 60 dB); S2: imperative auditory stimulus (frequency: 1500 Hz; duration: 50 ms; intensity: 60 dB). (from Mannarelli et al., 2023) [90].

**Table 1 brainsci-13-01683-t001:** Summary of the studies investigating the cerebellar contribution to attention, executive functions and timing using ERPs.

Cognitive Function Explored	Study	Subjects	NIBS	Stimulus Modality	Paradigm	Response Mode	Component Evoked	Results
Attention	Kremlacek, J. et al., 2011 [52]	n = 10 (SCA 2)	no	visual	oddball	motor	P3b	↑ latency
Attention	Adamaszek, M. et al., 2013 [50]	n = 1 (stroke)	no	visual and auditory	oddball	counting	EPN	absent
							LPP	↑ amplitude
Attention	Knolle, F. et al., 2013 [48]	n = 10 (cerebellar focal lesions) and n = 10 HS	no	auditory	auditory–motor paradigm	motor	N100	No suppression
Attention	Mannarelli, D. et al., 2015 [49]	n = 1 (stroke)	no	auditory	oddball	counting	P3b	↓ amplitude
Attention	Mannarelli, D. et al., 2016 [64]	n = 15 HS	tDCS	auditory	novelty P3	counting	N100	↓ latency and ↓ amplitude after cathodal tDCS
Attention	Bersani, F.S. et al., 2017 [62]	n = 42 (bipolar disorder)	tDCS	auditory	novelty P3	counting	P3b	↑ amplitude and ↓ latency after active tDCS
Attention	Houston, J.R. et al., 2018 [54]	n = 20 (CMI) and n = 20 HS	no	visual	dual task	motor	P3	↑ amplitude
Attention	Ruggiero, F. et al., 2019 [63]	n = 37 HS	tDCS	auditory	oddball	counting	N100	↑ amplitude after active tDCS
Attention	Tunc, S. et al., 2019 [51]	n = 25 (CA) and n = 30 HS	no	auditory	distraction paradigm	motor	P3a	↓ amplitude
							RON	↓ amplitude
							Pe	↓ amplitude
							MMN	normal
							ERN	normal
Attention	Rodríguez-Labrada, R. et al., 2019 [53]	n = 30 (SCA 2) and n = 33 HS	no	visual and auditory	oddball	motor	P3	↑ latency
Attention	Andrew, D. et al., 2020 [64]	n = 20 HS	cTBS	somatosensory	oddball	none	MMN	↓ amplitude after cTBS
Attention	Mauriello, C. et al., 2022 [59]	n = 23 ADHD vs. n = 23 HS	no	visual	delayed-face matching task	motor	P100	normal
							P2	↓ amplitude
							P3a	no change after active tDCS
							P300	no change after active tDCS
							N200	↓ amplitude after cathodal tDCS
							P3a	↓amplitude and =latency after cathodal tDCS
							P3b	↓ amplitude and =latency after cathodal tDCS
Executive functions	Bauer, L.O. et al., 2002 [75]	n = 66 (drug abusers) and n = 18 HS	no	visual	response competition task	motor	SP	↓ amplitude
Executive functions	Tanaka, H. et al., 2003 [79]	n = 13 (CCA) and n = 13 HC	no	visual	continuous performance task	motor	NoGo GFP	↑ latency and ↓ amplitude
Attention/ Executive functions	Paulus, K.S. et al., 2004 [45]	n = 1 (stroke)	no	auditory	oddball	counting	P3b	↓ amplitude
Executive functions	Mannarelli, D. et al., 2020 [81]	n = 16 HS	tDCS	auditory	Go/NoGo task	motor	NoGo N2	↑ latency and ↓ amplitude after cathodal tDCS
							P3	no change after cathodal tDCS
Executive functions	Yao J. et al., 2023 [82]	n = 36 HS	rTMS	visual	2-back working memory task	motor	N170	↑ amplitude after excitatory rTMS
							P300	↑ amplitude after excitatory rTMS
Executive functions	Saarikivi K. et al., 2023 [83]	n = 35 musicians and n = 28 controls	no	visual	set-shifting task	motor	P300	↑ amplitude
Timing	Moberget, T. et al., 2008 [87]	n = 7 (cerebellar degeneration) and n = 10 HS	no	auditory	MMN paradigm	none	MMN	↑ latency
Timing	Kotz, S.A. et al., 2014 [88]	n = 11 (cerebellar lesions) and n = 11 HS	no	auditory	oddball	counting	P3b	↓ amplitude
Timing	Mannarelli, D. et al., 2023 [90]	n = 16 HS	tDCS	auditory	CNV task	motor	CNV	↓ amplitude after cathodal tDCS (short and target intervals)

Note: SCA2: spinocerebellar ataxia type 2; HS: healthy subjects; CMI: type I Chiari malformation; CA: cerebellar ataxia; ADHD: attention deficit hyperactivity disorder; CCA: cerebellar cortical atrophy; NIBS: non-invasive brain stimulation; tDCS: transcranial direct current stimulation; cTBS: continuous theta burst stimulation; rTMS: repetitive transcranial magnetic stimulation; CNV: contingent negative variation; MMN: mismatch negativity; RON: reorienting negativity; ERN: error-related negativity; Pe: error-related positivity; NoGo GFP: NoGo global field power; SP: slow potential. ↓ indicates a reduction in amplitude or latency; = indicates no variation; ↑ indicates prolonged latency or greater amplitude.

## Data Availability

No new data were created or analyzed in this study. Data sharing is not applicable to this article.

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
