# Peer review of "Cerebellum’s Contribution to Attention, Executive Functions and Timing: Psychophysiological Evidence from Event-Related Potentials"

_brainsci, 2023, doi:10.3390/brainsci13121683_

Round 1
Reviewer 1 Report
Comments and Suggestions for Authors
This is a review articl. The review focused on data on attention, executive functions, and time processing obtained in healthy subjects and patients with varying clinical conditions, thus confirming the role of ERPs in understanding the role of cerebellum in cognition and exploring the potential diagnostic and therapeutic implications of ERP-based assessments in patients.
The review tried to listed a lot of publications to highlight that the contribution of erebellum between attention, executive functions, and time processing.
There are several aspects that needed to be improved.
This article serves as a review focusing on data regarding attention, executive functions, and time processing obtained from both healthy subjects and patients with various clinical conditions. Its primary aim is to confirm the role of event-related potentials (ERPs) in understanding the contribution of the cerebellum to cognition, while also exploring the potential diagnostic and therapeutic implications of ERP-based assessments in patients. In order to emphasize the significance of the cerebellum in attention, executive functions, and time processing, the review attempts to include an extensive list of publications.
Nevertheless, there are several areas that require refinement and improvement.
1. The scope of the title is too broad; the term "cognition" encompasses more perspectives than those mentioned, particularly attention, executive functions, and time processing.
2. There is a lack of enough references from the past five years.
3. The absence of a detailed section regarding limitations and future directions renders it an inadequate choice for a review.
4. Theoretical accounts are rare.
5. There is limited information on whether the cerebellum contributes to various cognitive tasks, such as various attentional tasks and various executive functions, and so on.
Reviewer 2 Report
Comments and Suggestions for Authors
The name sounds intriguing. The reader is primed to obtain information about the cognitive functions of the cerebellum from recordings of event-related potentials. The review covers three areas of psychophysiology of the human brain – attention, executive functions and time processing, studied according to the literature in healthy subjects and patients with various clinical conditions. The review is undoubtedly of interest to both specialized clinicians and neurophysiologists. In order to enhance the rating of this review, we can recommend paying attention to the following:
1. The authors got carried away with the use of the term ERP. In this case, information is mainly provided about the late components of evoked potentials. In this regard, in all research episodes it is necessary to replace ERP or add potentials of a specific modality to the description - visual, auditory, somatosensory, motor, magnetic (VEPs, AEPs, SEPs, etc.).
2. In addition, in research episodes, features of the stimuli used should be added. Even within specific paradigms (oddball, Go/NoGo, etc.) there can be significant differences.
3. There are unsuccessful expressions that require both replacement and decoding. Line 196 - pediatric population [62]. Are these patients? what age?
4. Line 209 – 210. Cerebellar involvement in executive functioning has been documented by experimental ERP studies, primarily in clinical populations. The term "populations" sounds like jargon in this context. It must be decrypted with the appropriate source. Moreover, as a rule, executive functions are studied in Parkinson's disease.
5. In the review, I strongly recommend borrowing the most interesting figures from the cited sources. For example, LORETA allows you to visualize the source of activity.
6. The review does not contain analytics. The authors list a series of studies related to this topic. This is fine. For example, a review will sound better if formed and described with analytics of psychophysiological states and the use of EPs of different modalities.
Reviewer 3 Report
Comments and Suggestions for Authors
This review article focuses on addressing the involvement of the cerebellum in selective cognitive functions by using ERPs and ERPs with noninvasive brain stimulations. However, certain issues need to be addressed before publication.
Do not use acronyms in the title.
On page 2, what does it imply “lobules I–IV-V”? You could directly say “lobules I-V”
As for better reader understanding, It would be better if you provide a graphical representation of how ERPs work/mechanism.
If it’s possible, you could add a table to notify the effectiveness of EPRs with other techniques like “improved or not improved after XXX days/weeks” with other published articles. For example “Article/Year/Technique/period/Condition”
I'd like to point out that recent findings have to be addressed. Like
“Bao H, Xie M, Huang Y, et al. Specificity in the processing of a subject's own name. Soc Cogn Affect Neurosci. 2023;18(1):nsad066. doi:10.1093/scan/nsad066
Proverbio AM, Tacchini M, Jiang K. What do you have in mind? ERP markers of visual and auditory imagery. Brain Cogn. 2023;166:105954. doi:10.1016/j.bandc.2023.105954”
Authors need to address the limitations/issues of the ERP techniques from a clinical perspective.
Round 2
Reviewer 1 Report
Comments and Suggestions for Authors
This paper have enhanced by the revision for a better version.
Reviewer 2 Report
Comments and Suggestions for Authors
I'm seeing a decent review now. In my opinion, in Fig. 1 you need to add the source to the signature.
Reviewer 3 Report
Comments and Suggestions for Authors
All appropriate changes were made in the revised manuscript. Henceforth, the manuscript can be accepted and I recommend the manuscript for publication